# Experimental and Numerical Investigation of CMT Wire and Arc Additive Manufacturing of 2205 Duplex Stainless Steel

Yuheng Yuan [1], Ruifeng Li [1,*], Xiaolin Bi [1], Jiayang Gu [2] and Chen Jiao [2]

1  School of Materials Science and Engineering, Jiangsu University of Science and Technology, Zhenjiang 212003, China
2  Marine Equipment and Technology Institute, Jiangsu University of Science and Technology, Zhenjiang 212003, China
*  Correspondence: li_ruifeng@just.edu.cn

**Abstract:** In this paper, the mechanical properties, microhardness and metallographic structure of 2205 duplex stainless steel by cold metal transfer (CMT) wire and arc additive manufacturing process are studied. The results show that the ultimate tensile strength, yield strength and elongation at break of reciprocating additive along building direction (BD) are 856.73 MPa, 710.5 MPa and 42.35%, respectively. In addition, the same direction motion (SDM) and reciprocating motion (RM) is selected as parameter variables in the experiment, and the finite element model is established by ABAQUS software, and the temperature and residual stress field of the additive forming at different paths are tested and simulated. Firstly, the accuracy of the selected finite element model was verified by comparing the experimental results from the simulation results to the macroscopic morphology of the cross-section of the single-pass additive specimen. The numerical simulation results show that due to the difference of the additive scanning paths, the distribution of the temperature field has a large difference, and with the increase of the deposited layer, the heat accumulation of the SDM additive is larger than that of the RM, so that the end collapses of the SDM additive will occur in the actual additive specimen. By simulating and comparing the equivalent stress distribution of different paths, the equivalent stress distribution of SDM and RM is approximately the same in the vertical direction, and the minimum of equivalent stress appears at the bottom of the deposition layers, about 116.5 MPa, and the maximum of equivalent stress appears at 8 mm from the top, about 348 MPa.

**Keywords:** CMT wire and arc additive manufacturing; 2205 duplex stainless steel; different scan paths; temperature field; stress field

## 1. Introduction

With the increasing requirements for metal products, the application of additive manufacturing technology in the field of metallic materials has gradually broadened and has gained increasing attention worldwide in the fields of automotive manufacturing, aerospace, nuclear power, and shipbuilding [1–5]. Additive manufacturing (AM) is a technology that reduces costs by reducing material waste and saving time [6,7]. In addition, AM technology does not require traditional tools and fixtures and multi-pass machining processes to quickly and accurately manufacture parts of any complex shape on a single machine. Solved many complex structural parts forming, and greatly reduce the processing procedures, shorten the processing cycle [8]. Wire and arc additive manufacturing (WAAM) is a promising technology that uses an arc as a heat source of the deposition of metal wires. Although the accuracy of WAAM is lower than that of powder-based AM, the technique is well suited for the production of medium complexity and large components [9,10].

WAAM uses the arc generated by different welding methods as a heat source to melt the metal wire, which is stacked layer by layer according to pre-defined parameters and paths to manufacture the desired metal parts from the bottom up [11]. This technology

has the characteristics of fast forming speed, low equipment cost, high material utilization rate, and good overall performance of the formed parts, which is receiving more and more attention research scholars. CMT technology is developed on the basis of short-circuit transfer. However, unlike ordinary GMAW. The welding wire not only has the movement of forward wire feeding, but also has the action of backward pumping. The welding process is: arc burning, welding to wire sent forward, until the formation of droplet short circuit, at this moment, the wire feeding speed reversed, the wire pulled back, then the current and voltage is almost zero. After the next open circuit is formed, the arc is re-ignited, the welding wire is sent forward again, and the droplet transfer begins again. Among them, CMT technology has become one of the main heat sources for WAAM because of its low heat input, low spatter, and high manipulability [12,13].

Nowadays, duplex stainless steels are widely used in industry. Composite plates of 2205 duplex stainless steel and Q345C have also been used in desilting pipes and deep flood discharge holes of bridge works. The potential application background of 2205 duplex stainless steel additive manufacturing in petroleum and natural gas engineering, marine field, chemical industry and other fields. The use of AM can greatly shorten the production time and reduce costs. It is of great significance to analyze and discuss the arc additive manufacturing technology of 2205 duplex stainless steel. Hengsbach et al. [14] investigated the law of influence of selective laser melting (SLM) forming process parameters on the structure of UNS S31803 duplex stainless steel. The results showed that too fast a cool rate resulted in too low austenite content, causing an imbalance in the ratio of the two phases and creating a high dislocation density within the specimen. Wang et al. [15] investigated the effect of varying the process parameters (line energy density input) on the microstructure and mechanical properties of multilayer single-pass specimens of 304 L austenitic stainless steel fabricated by laser metal deposition (LMD), and the results showed that by choosing a lower line energy density can lead to a more uniform microstructure distribution of the specimen and greatly enhance the mechanical properties of the material. Hosseini et al. [16] compared the forming morphology and microstructure of 2205 duplex stainless steel at high heat input, high interlayer temperature and low heat input, low interlayer temperature based on Gas metal arc welding additive manufacturing (GMAW-AM). The results showed that the morphology and austenite contents of the formed parts were more similar under both thermal circulation conditions, but the austenite grain size was finer in the low interlayer temperature specimens, while the precipitated phases were more in the high interlayer temperature.

While WAAM has tremendous advantages in AM processes, the heat input of WAAM is relatively high compared to LMD [17]. In the process of additive manufacturing, due to local uneven heating, internal stress will be generated inside the component, resulting in deformation of the component [18]. Higher residual stresses can cause significant distortion and deformation, leading to failure of the workpiece, which is more pronounced for thin plates [19]. Therefore, it is crucial to better control the effects of residual stresses and deformations.

Finite element analysis is heavily used in the production of WAAM processes as a stable, cheap and reliable tool. Luo and Zhao et al. [20] reviewed the existing finite element analysis based thermodynamic models and summarized the advantages and disadvantages of these models in 2018. Bertini et al. [21] analyzed the laser powder bed fusion (L-PBF) residual stresses in 2019. Cook et al. [22] used computational fluid dynamics (CFD) models to simulate the L-PBF process. Li et al. [23] investigated the geometric optimization of the end transverse extension path strategy by the laser and CMT composite additive manufacturing.

Many researchers have studied the effects of different paths on the temperature and stress fields. Zhou et al. [24] analyzed the temperature field and stress field by studying different paths in multilayer single-pass. Zhao et al. [25] studied the evolution of thermal stresses and the effect of residual stresses in a multilayer single-pass additive process. Somashekara et al. [26] investigated the effect of zone filler path on residual stresses during

weld deposition using a combination of finite element analysis and and experimental methods. Sun et al. [27] investigated the effects of zig-zag, raster, alternate-line, out–in spiral, in–out spiral and Hilbert paths on the temperature and stress fields using finite element analysis.

At present, CMT wire and arc additive manufacturing is a very novel manufacturing process. Having a high heat input leads to the generation of large residual stresses, which is one of the main challenges to be addressed. The above-mentioned studies show that finite element analysis is an efficient, convenient and practical method in order to improve our breakthroughs in the field of additive manufacturing. It can solve the problems encountered in real-world AM through computer simulation. In this paper, ABAQUS finite element analysis software will be used. The distribution of temperature fields and stress fields in 2205 duplex stainless steels during the CMT process will be numerically simulated. This explains the heat accumulation and thus collapse process in the actual additive manufacturing process for SDM additive, while RM additive are better formed.

## 2. Methods and Experimental Design

### 2.1. Materials and Instruments

Q235A mild carbon structural steel (Jiangsu Ruixiang Metal Materials Co., Ltd., Taizhou, China) was used as the base plate. The thickness of the base plate is 10 mm. The wire used in the test is the Swedish SANDVIK company (Twicken, Sweden) to provide ER2209 duplex stainless steel wire. Wire specifications $\phi$ = 1.6 mm. The schematic diagram of its AM is shown in Figure 1a. At room temperature, the SANS universal tensile testing machine (Tianjin Meters Testing Machine Factory, Tianjin, China) is used as the test equipment, the tensile rate is 0.5 mm/min, and the size of the tensile specimen is shown in Figure 1b. The chemical composition and main physical and mechanical properties of the base plate and wire for AM are shown in Tables 1 and 2.

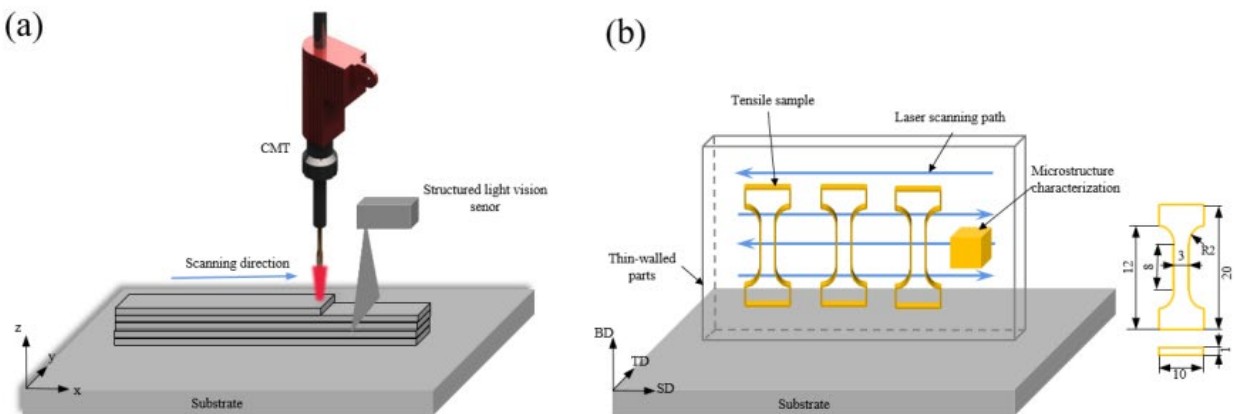

**Figure 1.** WAAM process. (**a**) Schematic diagram of CMT wire and arc additive manufacturing. (**b**) Extraction and size of tensile samples.

**Table 1.** Chemical composition of base metal and welding wire (wt.%).

| Materials | C | Si | Mn | P | S | Cr | Ni | Mo | N | Fe |
|---|---|---|---|---|---|---|---|---|---|---|
| Q235A | 0.14~0.22 | ≤0.35 | 0.30~0.65 | ≤0.045 | ≤0.050 | - | - | - | - | Bal. |
| ER2209 | 0.013 | 0.49 | 1.54 | 0.018 | 0.007 | 22.92 | 8.6 | 3.2 | 0.17 | Bal. |

**Table 2.** Main physical and mechanical properties of base metal and welding wire.

| Materials | $\rho$ (g/cm$^3$) | Modulus of Elasticity E (GPa) | Coefficient of Linear Expansion $\alpha$ (10$^{-5}$/°C) | Yield Strength Rp$_{0.2}$ (MPa) | Tensile Strength Rm (MPa) | Elongation $\delta$ (%) |
|---|---|---|---|---|---|---|
| Q235A | 7.85 | 200~210 | 12 | 235 | 370~550 | $\geq$25 |
| ER2209 | 7.98 | 190~210 | 13.7 | 450 | 620 | $\geq$25 |

The equipment used for the experiment consists mainly of the robot part and the CMT welding system part. The robot used in this test is the IRB 1600-6/1.2 (ABB Robot Co., Ltd., Zhuhai, China) six-axis robot manufactured by the ABB. This robot can execute the action strictly according to the predetermined program under the high-speed continuous working condition, so that WAAM can be guaranteed in the forming accuracy. The controller used with the IRB 1600-6/1.2 welding robot is the IRC5 (ABB Robot Co., Ltd., Zhuhai, China). The IRC5 model has excellent flexibility, accuracy and popularity, thus ensuring that the robot can strictly follow the established trajectory during WAAM process. The welding system used in this test is part of the CMT Advanced 4000 welding machines manufactured by Fronius Austria (Beijing, China). The type of welding powers supply is TPS4000 (ABB Robot Co., Ltd., Zhuhai, China) digital welding power supply, which can provide three welding modes including direct current welding, pulse welding and CMT welding, through the external RCU5000I (ABB Robot Co., Ltd., Zhuhai, China) remote control can be simple and convenient to manipulate the welding process. At the same time, through the cooperation of the power supply and the built-in software, the welding system achieves the unified adjustment of the welding current, arc voltage and wire feed speed, i.e., after selecting the welding method, wire material, wire diameter, protector type and shielding gas parameters on the welding work panel, the welding current and arc voltage can be adjusted simultaneously by adjusting the wire feed speed. The cooling system selected for the test was the FK4000R (ABB Robot Co., Ltd., Zhuhai, China), a water tank with efficient cooling capacity that allows for standard temperature control and water flow control.

### 2.2. CMT Wire and Arc Additive Manufacturing

In this paper, two different additive scanning paths, SDM additive and RM additive, are used for multiplayer single-pass research. The SDM additive means that the additive scanning direction is along the same direction. The RM additive means that the additive direction of two adjacent layers is opposite. Other conditions remain unchanged, the scanning speed is set to 6 mm/s. The wire feeding speeds to 60 mm/s, the interlayer cooling time to 60 s, and the number of deposited layers to 10. The macroscopic shapes of the formed parts obtained according to the two additive paths are shown in Figure 2. For the SDM path additive, the scanning direction of the 10 layers is the same. The expansion at the striking arc and the tilt at the extinguishing arc are accumulated layer by layer, resulting in a significant height difference between the starting point and the end point of the trajectory, while for the RM path additive, the scanning direction is exchanged layer by layer, and the expansion at the striking arc point and the tilt at the extinguishing arc point compensate each other, and the upper surfaces appearance of the part is flat.

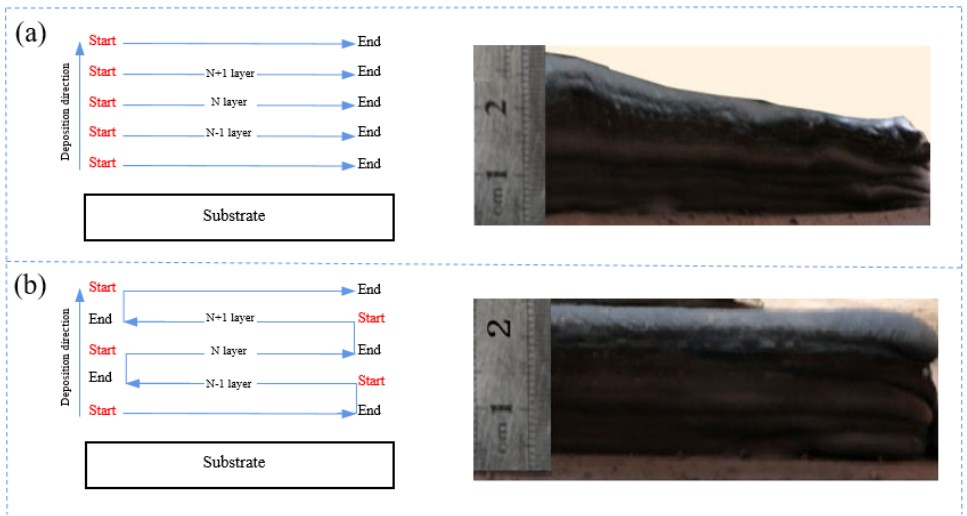

**Figure 2.** Additive paths and macroscopic morphology. (**a**) SDM scanning path and macroscopic morphology. (**b**) RM scanning path and macroscopic morphology.

## 3. Finite Element Analysis

### 3.1. Finite Element Model

In order to predict the changes in temperature and residual stresses during the AM process, finite element analysis will be performed using ABAQUS (6.14, 2014, ABAQUS, Paris, France). The size of the finite element model is modeled according to the measured size of the actual additive. The established geometric models and finite element meshing are shown in Figure 3. Where the size of the substrate is 200 mm × 100 mm × 10 mm, the deposit layer is 180 mm × 7 mm × 30 mm, and each layer is 180 mm × 7 mm × 3 mm. ABAQUS finite element analysis software can predict the temperature, residual stresses and deformation during AM process, which can be justified for realistic tests. Sparse meshes in the process of finite element modeling can have an impact on the accuracy of the simulation. The fine-grained mesh increases the computation time significantly. Therefore, it is very important to perform finite element analysis to divide the appropriate mesh. In order to obtain a more accurate mathematical model, this simulation will use temperature-displacement coupling for finite element analysis. The type of mesh used is an 8-node thermally coupled hexahedral cell with three-way linear displacement, three-way linear temperature (C3D8T) [25]. A total of 19712 meshes were divided by inspection of the mesh [28]. Figure 4 represents the temperature-dependent material properties of 2205 duplex stainless steel.

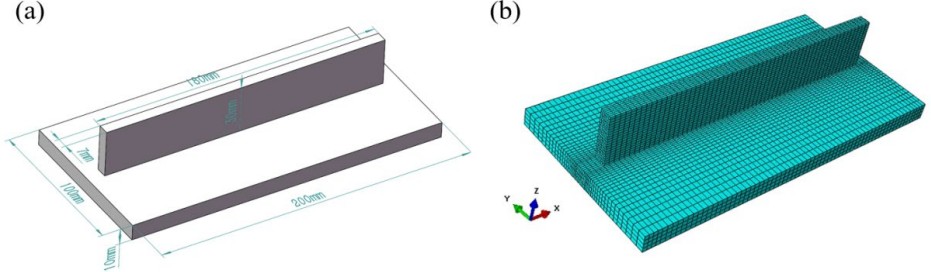

**Figure 3.** Geometric model. (**a**) Geometric model dimensions. (**b**) Finite element analysis divides the mesh.

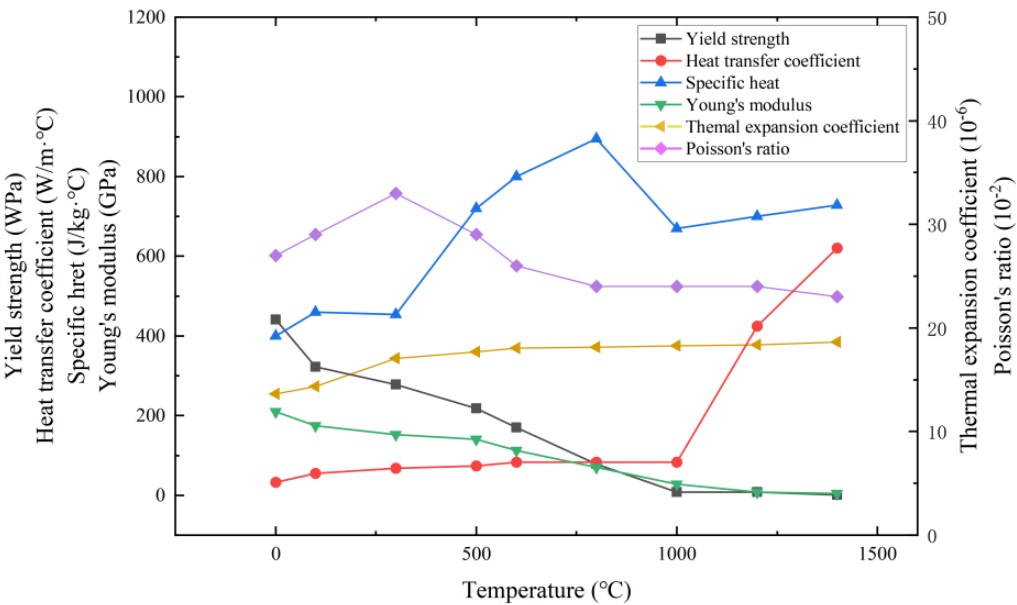

**Figure 4.** Material properties of 2205 duplex stainless steel with temperature.

### 3.2. Selection of Heat Source

In the process of CMT wire and arc additive manufacturing, a double ellipsoid heat source model [29–31] will be used, as shown in Figure 5. This model defines a heat through two ellipsoids. The dimensions of the two ellipsoids are determined by four parameters ($a_1$, $a_2$, b and c) [32], as shown in Table 3. The heat flux of the node with coordinates (x, y, z) is different depending on its division into two parts, front and back, with the expression [28,33]:

$$q_1(x, y, z) = \frac{6\sqrt{3}f_1 Q}{\pi a_1 bc\sqrt{\pi}} \exp\left[-\frac{3(x - x_0 - d)^2}{a_1^2} - \frac{3(y - y_0)^2}{b^2} - \frac{3(z - z_0)^2}{c^2}\right] \tag{1}$$

$$q_2(x, y, z) = \frac{6\sqrt{3}f_2 Q}{\pi a_2 bc\sqrt{\pi}} \exp\left[-\frac{3(x - x_0 - d)^2}{a_2^2} - \frac{3(y - y_0)^2}{b^2} - \frac{3(z - z_0)^2}{c^2}\right] \tag{2}$$

$$f_1 + f_2 = 2 \tag{3}$$

$$f_1 = \frac{2a_1}{a_1 + a_2}, \ f_2 = \frac{2a_2}{a_1 + a_2} \tag{4}$$

where $Q = \eta UI$, $\eta$ is the heat efficiency, U is the welding voltage (V), I is the welding current (A); a, b, c are ellipsoid shape parameters; $(x_0, y_0, z_0)$ is the initial coordinate of the heat source; d is the moving distance of heat source; $f_1$, $f_2$ are the front and rear ellipsoid heat distribution functions.

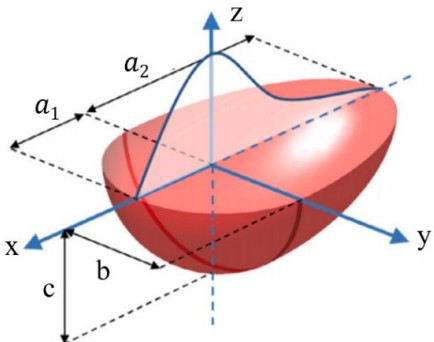

**Figure 5.** Double ellipsoid heat source model.

**Table 3.** Double ellipsoid heat source parameters.

| $a_1$ (mm) | $a_2$ (mm) | b (mm) | c (mm) |
| --- | --- | --- | --- |
| 3 | 6 | 4 | 4 |

*3.3. Heat Conduction Analysis*

The heat conduction process obeys the Fourier heat conduction law. The content is that the heat flux density q* [J/(mm²·s)] on the isothermal surface of the object is proportional to the negative temperature gradient perpendicular to the isothermal surface, and is proportional to the thermal conductivity λ. The specific expression is:

$$q^* = -\lambda \frac{\partial T}{\partial n} \tag{5}$$

where T is the temperature; n is the normal vector of isothermal surface at this point, partial derivative is temperature gradient; λ is the thermal conductivity.

The heat conduction equation can also be written into a more general expression:

$$q^* = -\lambda \nabla T = -\lambda \left( i\frac{\partial T}{\partial x} + j\frac{\partial T}{\partial y} + k\frac{\partial T}{\partial z} \right) \tag{6}$$

Here, $\nabla$ is a three-dimensional inverted triangle operator; T (x, y, z) is the scalar temperature field. The heat flux density vector can be decomposed into several components. In the rectangular coordinate system, the general expression of q* is:

$$q^* = iq_x^* + jq_y^* + kq_z^* \tag{7}$$

The Formula (6) is available:

$$q_x^* = -\lambda \frac{\partial T}{\partial x}; \ q_y^* = -\lambda \frac{\partial T}{\partial y}; \ q_z^* = -\lambda \frac{\partial T}{\partial z} \tag{8}$$

Each of the expressions above illustrates the relationship between the heat flux through a surface and the temperature gradient in the normal direction of the surface. This means that the medium in which heat conduction occurs in formula 6 is anisotropic [34–36].

*3.4. Boundary Condition*

In the process of additive manufacturing simulation, the surface heat transfer coefficient and the ambient temperature are set to belong to the third boundary condition. Before the start of the simulation, the ambient temperature is set to 25 °C, and the surface heat transfer is uniform. In the simulation, the surrounding of the bottom plate is fixed. During the cooling process, the constraints are released.

## 4. Results and Discussion

*4.1. Experimental Result Analysis*

### 4.1.1. Mechanical Properties

Micro Vickers hardness of the sample was measured by HXS-1000AC (Shanghai Optical Instrument Import and Export Co., Ltd, Shanghai, China) semi-automatic microhardness tester. The loading load is 0.1 kg. The holding time is 15 s. Take 10 points on the BD, every two points 50 μm distance, measured microhardness values are shown in Figure 6a. The microhardness of 2205 duplex stainless steel is around 235.19 $HV_{0.1}$. The tensile test used CMT4303 (Tianjin Meters Testing Machine Factory, Tianjin, China) microcomputer control electronic universal testing machine. The room temperature tensile test results are shown in Figure 6b. The results show that the ultimate tensile strength, yield strength and elongation at break of reciprocating additive along BD are 856.73 MPa, 710.5 MPa and 42.35%, respec-

tively. The SEM fracture scanning along the BD tensile sample is shown in Figure 6c,d. The microstructure of fracture is dimple, which belongs to ductile fracture.

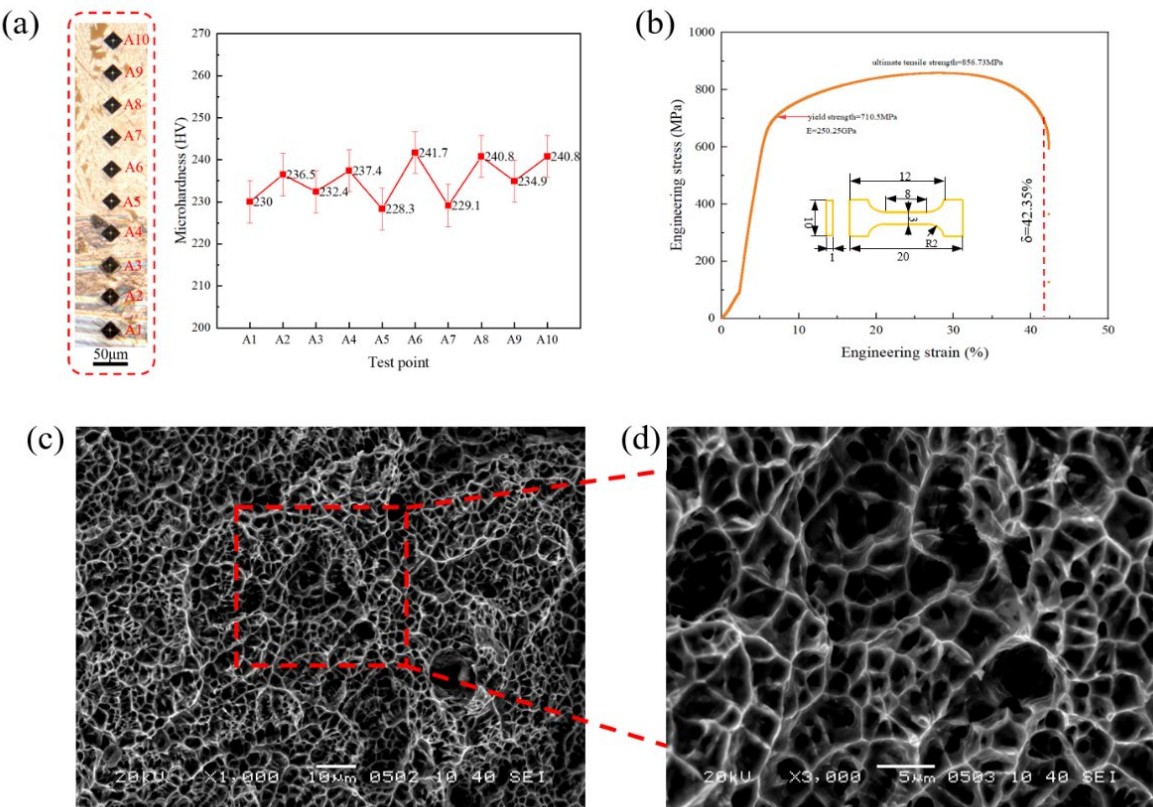

**Figure 6.** Mechanical properties. (**a**) Microhardness test. (**b**) Tensile test. (**c,d**) SEM fracture morphology diagram.

### 4.1.2. Microstructure Evolution

In the process of multi-layer single-pass WAAM, with the increase of the height of the deposited layer, the heat dissipation conditions of the molten pool begin to gradually deteriorate and the heat accumulation gradually increases. The molten pool will be subjected to a complex thermal cycle process, thus affecting the microstructure of 2205 duplex stainless steel. By observing the multilayer single-pass samples along the BD, the microstructures at different positions can be observed as shown Figure 7. In the process of additive manufacturing, the grain growth size is affected by cooling rate and undercooling. At the junction of the substrate, the cooling rate is the fastest. It can be seen that the bottom structure of the formed part is mainly dendritic and feathery austenite, Widmanstatten austenite is arranged in parallel, and the amorphous grain boundary austenite content is more. It can be seen that the main microstructure in the middle is columnar dendrites and coarse Widmanstatten austenite. The microstructure began to change from fine dendrite to columnar dendrite, and the grain size increased. This is because with the repeated heating of the sample by the welding torch, the heat in the molten pool accumulates and is farther and farther away from the substrate. It can only rely on convection with air to dissipate heat, resulting in poor heat dissipation conditions in the molten pool. At the top of the formed part, the structure is coarse Widmanstatten austenite and columnar equiaxed grains. The austenite size at the top is larger than that at the middle, because the heat dissipation condition at the top is the worst and the temperature gradient is the smallest.

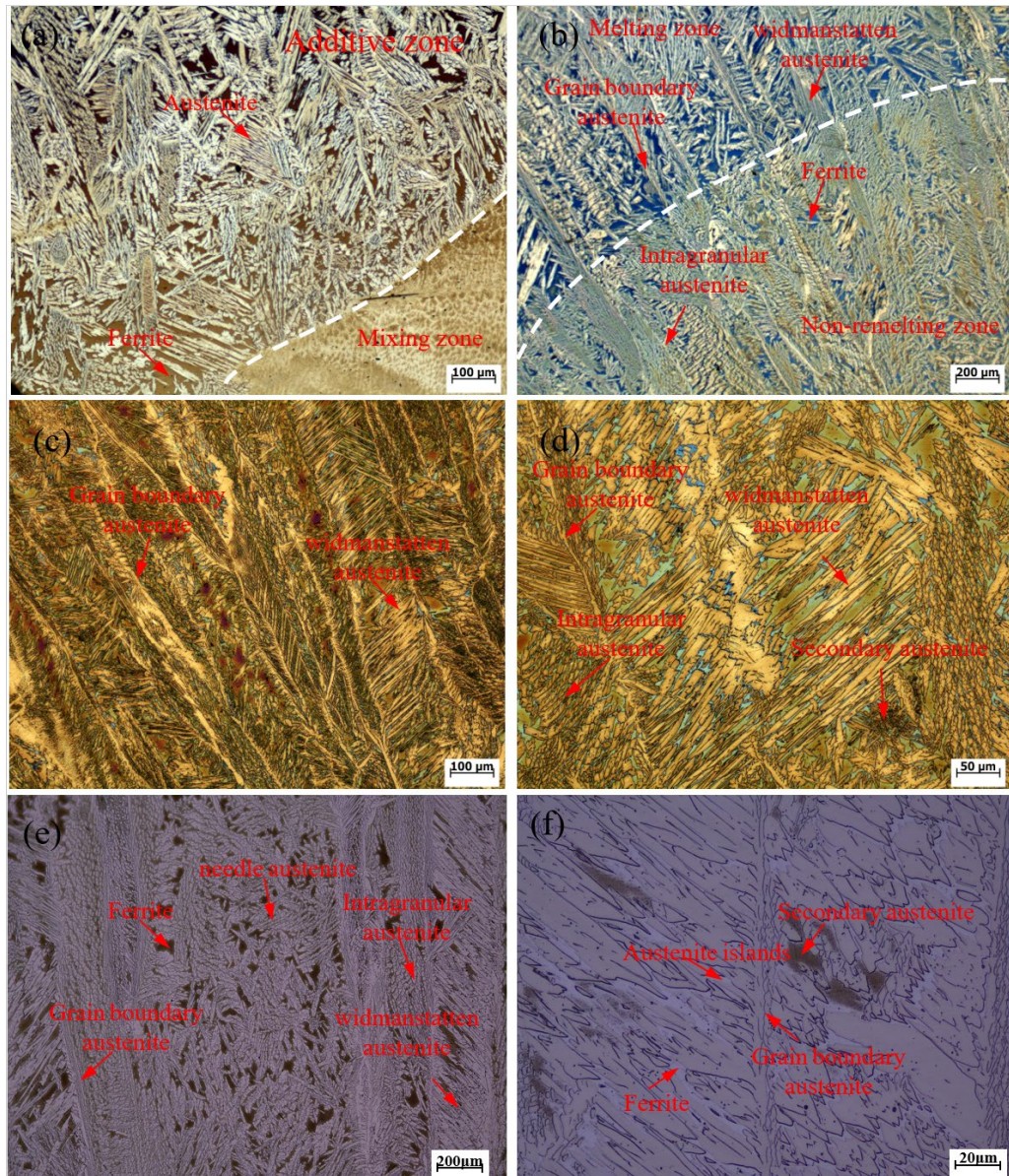

**Figure 7.** Microstructure of 2205 duplex stainless steel by WAAM. (**a**,**b**) Microstructure at the bottom along BD direction. (**c**,**d**) Microstructure in the middle along BD direction. (**e**,**f**) Upper microstructure along BD direction.

*4.2. Simulation Result Analysis*

4.2.1. Calibration of the Finite Element Model

In this paper, the heat source model was first calibrated and the deposition of mono-layer single-pass was carried out by the CMT wire and arc additive manufacturing process, using the same process parameters as in Section 2. After the monolayer single-pass AM testing, the morphology of the monolayer single-pass cross-section was observed under an inverted optical microscope (IOM). The temperature field of the monolayer single-pass was simulated by ABAQUS software, as shown in Figure 8a. The simulated temperature field is compared with the actual single layer additive specimen formation, as shown in Figure 8b. The simulated morphology is similar to the actual melt pool morphology. This means that the actual heat input is comparable to the simulated heat input. It can be used as a heat input for subsequent simulations of multilayer single-pass additions.

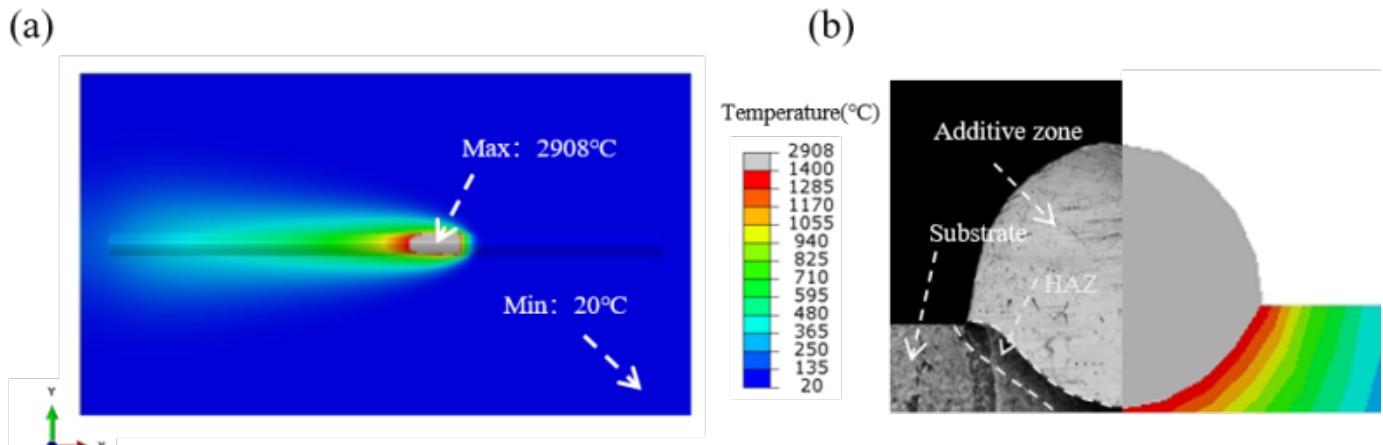

**Figure 8.** Calibration of the heat source model. (**a**) Temperature field distribution of a monolayer single pass. (**b**) Comparison of the cross-sectional shape of the actual additive formation and the simulated results.

### 4.2.2. Temperature Field Analysis

To better analyze the thermal behavior of different points during the deposition process. The thermal cycling curves and temperature fields of the midpoints of the 1st, 5th and 10th layers were extracted from the model for analysis [37]. Figure 9 shows the temperature field distribution contour at different midpoints under different paths. The distribution of the contour shows that the shape of the melt pool is quite stable during the deposition. As shown in Figure 9a, the center temperature of the melt pool of the 1st layer of the SDM additive reached 2899 °C, the center temperature of the melt pool of the 5th layer reached 3004 °C, and the center temperature of the melt pool of the 10th layer reached 3315 °C. As shown in Figure 9b, the center temperature of the melt pool of the 1st layer of the RM additive reached 2899 °C, the center temperature of the melt pool of the 5th layer reached 2994 °C, and the center temperature of the melt pool of the 10th layer reached 3114 °C.

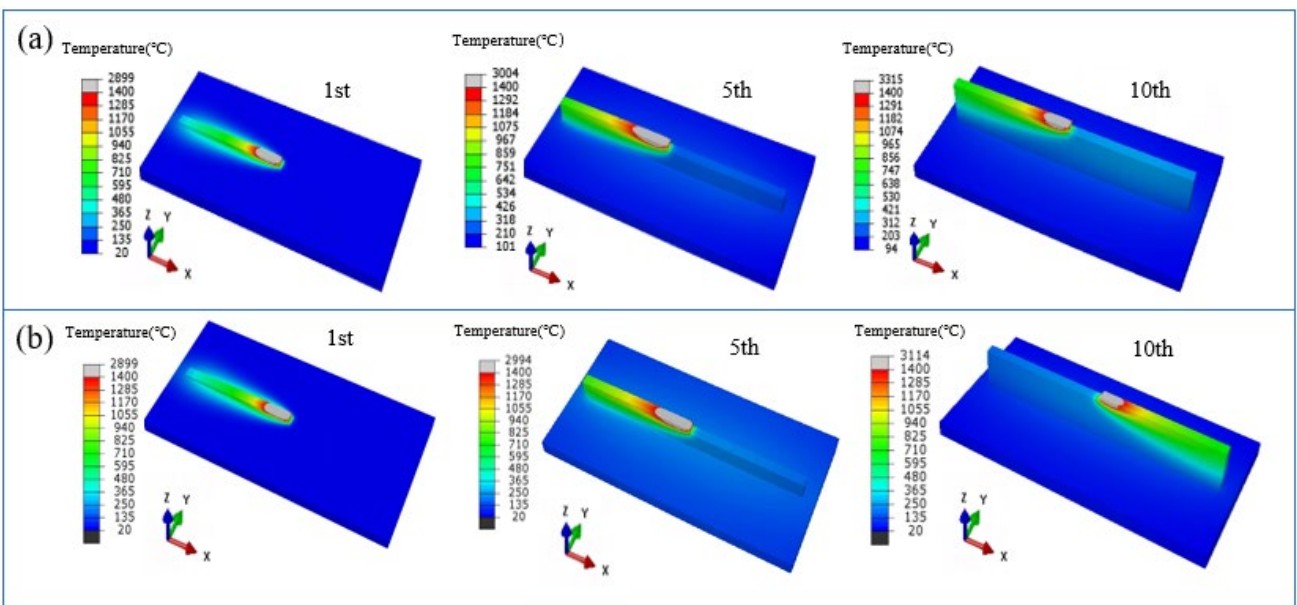

**Figure 9.** Temperature field distribution at different midpoints for different paths. (**a**) SMD scan path. (**b**) RM scan path.

Figure 10 shows the thermal cycling curves for different midpoints under different paths. Comparing the thermal cycling curves of the SDM and RM at the midpoints of different layers. The temperature changes at the midpoint are very similar because the time for the heat source to move to the midpoint of any layer is the same in the two different paths. During the deposition process, when the heat source reaches the midpoint of the first layer, the temperature at the midpoint peaks at this point, and when layer 2 is deposited, the heat source of layer 2 has a remelting effect on the first layer, causing the peak of the thermal cycling curves at this point to exceed the melting point temperature when deposition of layer 2 is performed. By observing the thermal cycling curves of layers 5 and 10, it can be seen that the peak temperature of the thermal cycling curves is higher compared to the peak temperature of the previous layer as the deposited layer increases. Therefore, by observing the temperature field distribution contour and thermal cycling curves at different midpoints under different paths, heat accumulation occurs as the deposition layer increases.

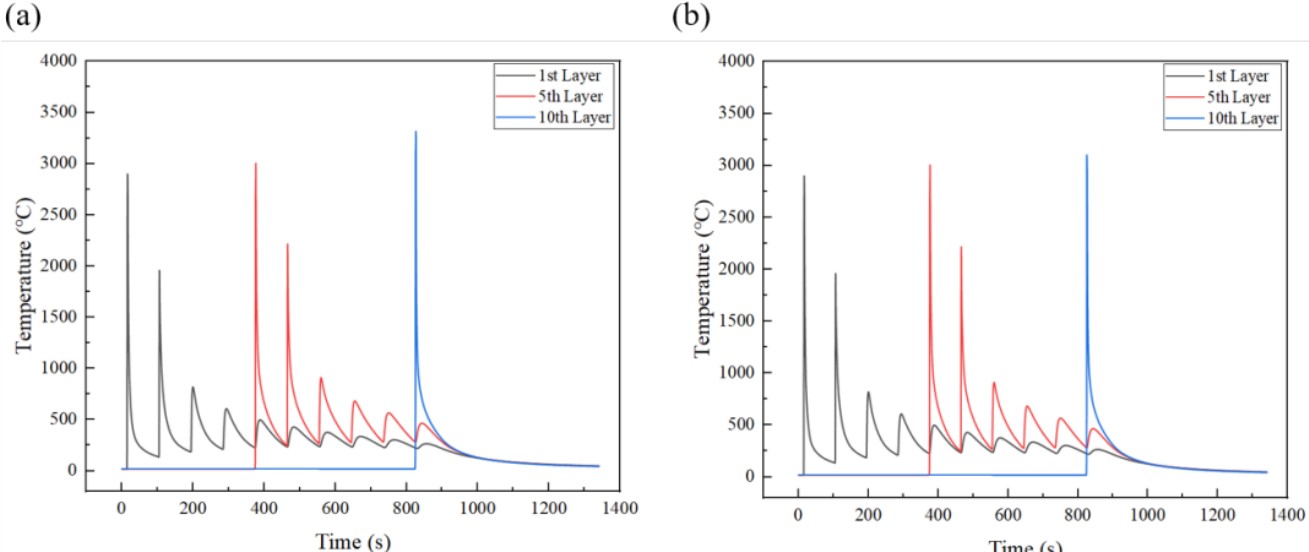

**Figure 10.** Thermal cycling curves at different midpoints for different paths. (**a**) SMD scan path. (**b**) RM scan path.

When the heat sources move in a different way, the resulting temperature distribution inside the workpiece will also be different. When the heat source moves to the middle of the 10th layer, the temperature distribution of the left and right two-end points along the height direction is, respectively, as shown in Figure 11. The temperature of the arc leading end is lower than that of the arc extinguishing end. Comparing the temperature distribution between the arc-striking end and the arc-extinguishing end, it is found that the temperature of SDM is higher than that of RM at the arc-striking end, while the temperature distribution of SDM is consistent with that of RM at the arc-striking end. Therefore, in the process of SDM and RM additive manufacturing, SDM additive manufacturing will accumulate more heat, resulting in the process of one end collapse.

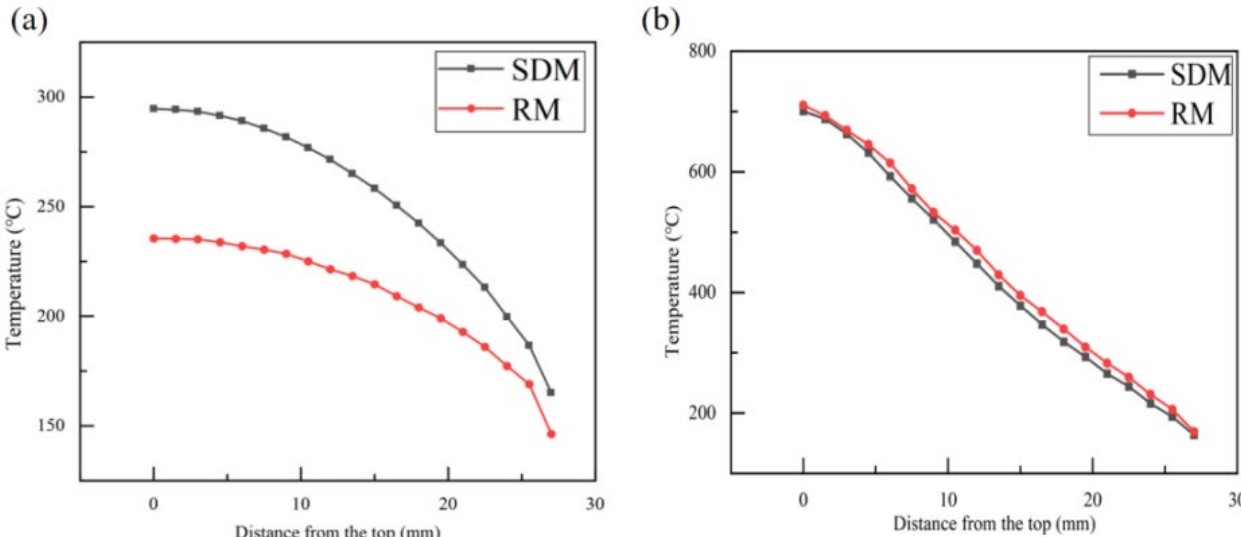

**Figure 11.** Temperature distribution along the height direction at the left and right end points of the last layer. (**a**) Arc-extinguished end. (**b**) Arc-striking end.

### 4.2.3. Stress Field Analysis

During the AM process, different scanning paths have different effects on the work piece and different stress distributions. In order to compare the distribution of stresses on the work piece by different paths. The equivalent stress distributions for different paths of additions to the 1st and 5th layer when cool about 60 s and the equivalent stress cloud distribution of the 10th layer when cool about 500 s were selected and are shown in Figure 12. Since the 1st layer SDM and RM paths are the same, the distribution of stresses is approximately the same. With the increase in deposition layers, the top stress of the deposition layer is higher than the bottom. When the deposited layer reached the 10th layer and subsequently cool about 500s, the maximum stress of the SDM reached 521.8 MPa, while the maximum stresses of the RM was 487.2 MPa. From the distribution of the contour, the stress is smaller at the lowest end of the deposition layer, and the stress increases and then decreases nearer to the upper end, and there is a concentration of stress at the two ends of the deposition layer.

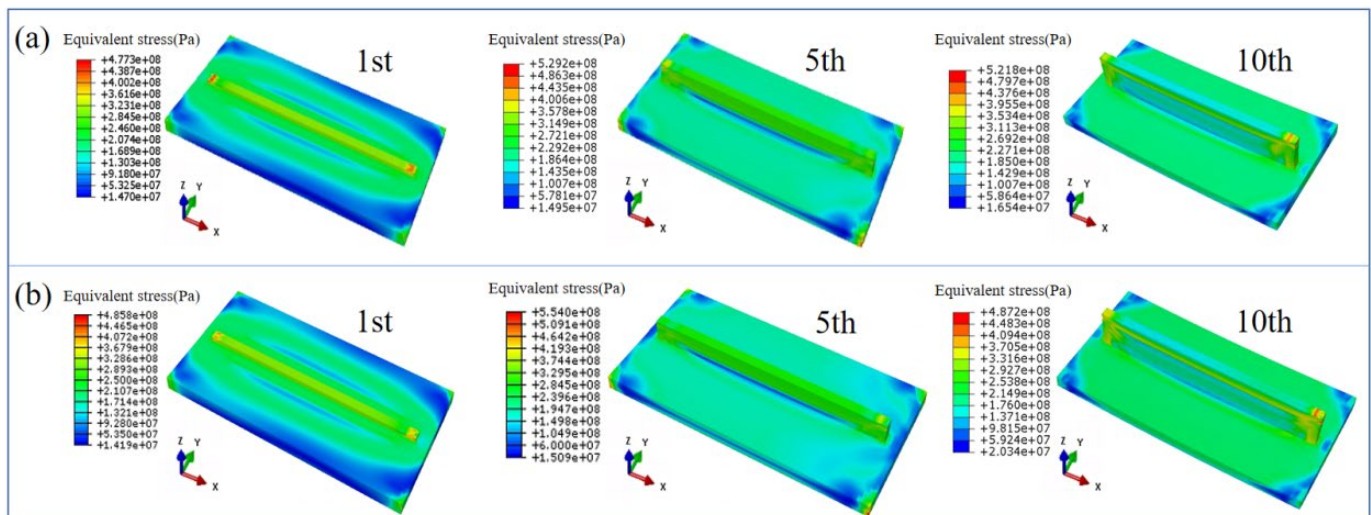

**Figure 12.** Contour of the equivalent stress distribution for different paths. (**a**) SDM. (**b**) RM.

When the deposition is completed and cool to 500s, the residual stresses along the AB path to the different paths are shown in Figure 13. Figure 13a shows the longitudinal

stresses as tensile stresses for both the SDM and RM. The longitudinal tensile stresses along the ends of the AB path are large and can reach a maximum of 130.5 MPa. The stress in the middle is smaller, with an average stress of about 25 MPa. From Figure 13b, it can be seen that the transverse stresses in different paths are compressive stresses. The compressive stresses in SDM are greater than those in RM due to the fact that more heat accumulates in SDM, making the stresses greater.

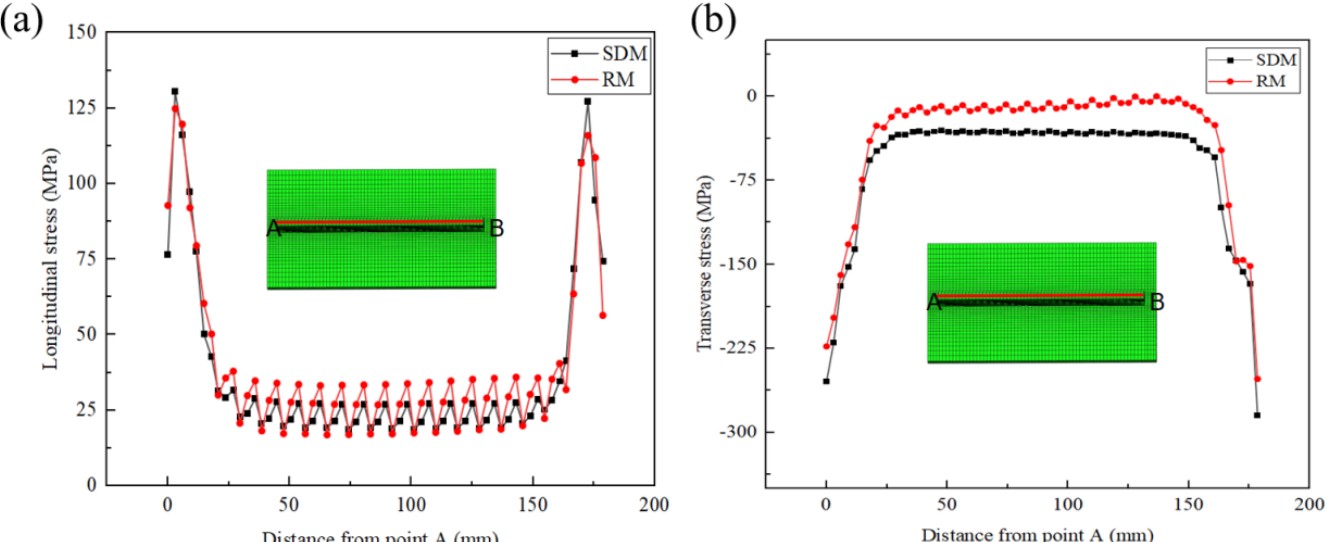

**Figure 13.** The residual stress distribution along the AB path under different paths. (**a**) Longitudinal stress. (**b**) Transverse stress.

Figure 14 shows the distribution of equivalent stress along the CD path. By comparing the equivalent stress distributions of SDM and RM, the equivalent stress distributions are approximately equal. The minimum equivalent stress near the bottom of the deposition layer is 116.5 MPa, and the maximum equivalent stress at 8 mm from point C is 348 MPa. Figure 15 shows the residual stress distribution along the CD path for different paths. From Figure 15a, it can be seen that the longitudinal stress distribution of the SDM and RM along the CD path near the C end is approximately the same, and the further away from the C point, the greater the longitudinal residual stress of the SDM. The closer to the point D, the greater the longitudinal stress of the SDM than the longitudinal stress of the RM. Figure 15b shows the transverse stress distribution along the path of CD, the tensile stress is mainly concentrated near point C. The compressive stresses are mainly near point D.

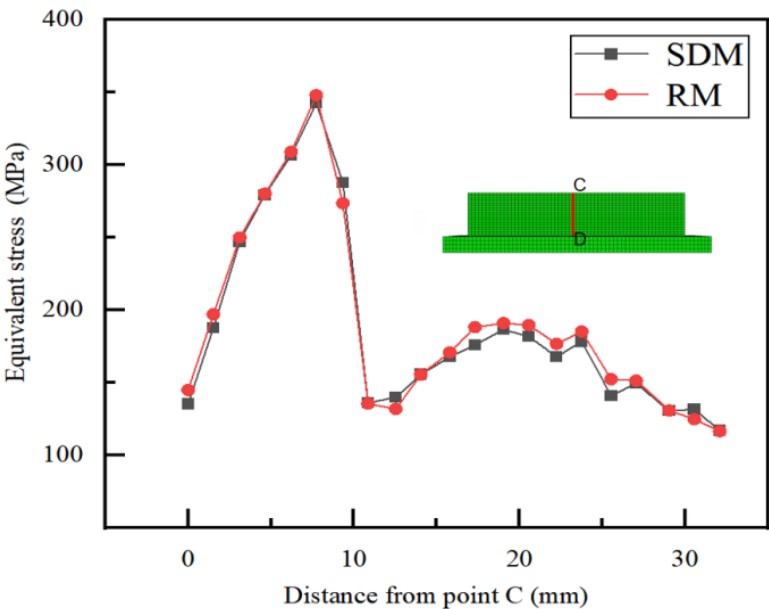

**Figure 14.** Equivalent stress distribution along the CD path.

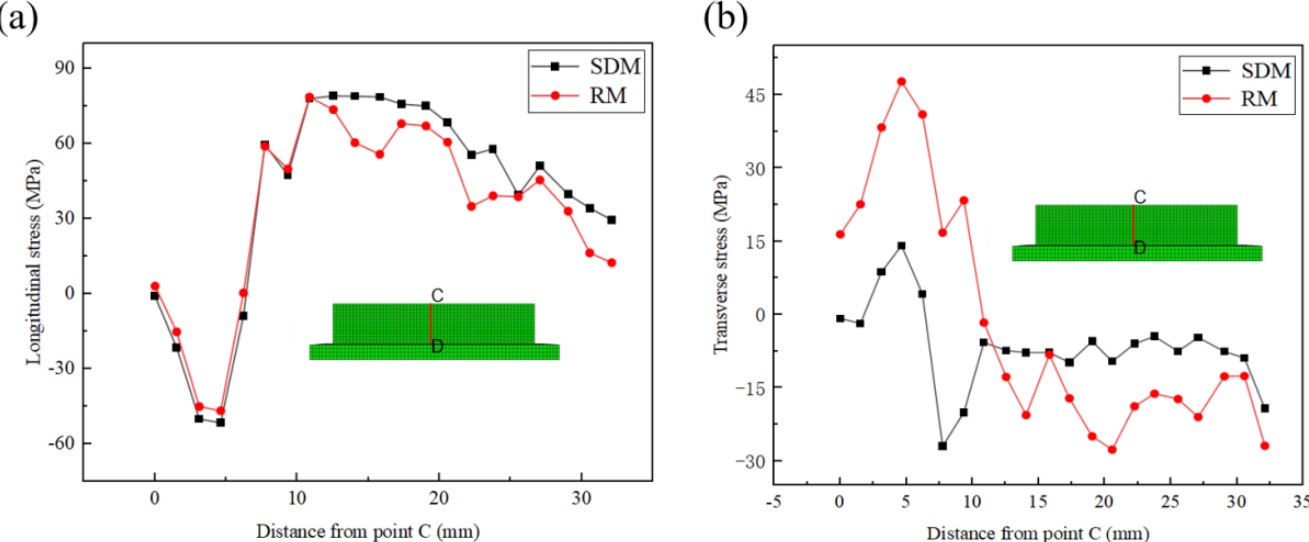

**Figure 15.** Residual stress distribution along the CD path under different paths. (**a**) Longitudinal stresses. (**b**) Transverse stress.

**5. Conclusions**

In this paper, the additive formation of 2205 duplex stainless steel based on CMT by means of experiments and finite element simulation, and the main conclusions are as follows.

1. The microhardness of 2205 duplex stainless steel is about 235.19 $HV_{0.1}$. The ultimate tensile strength, yield strength and elongation at break of reciprocating additive along BD are 856.73 MPa, 710.5 MPa and 42.35%, respectively. The microstructure of 2205 duplex stainless steel is mainly needle austenite, intragranular austenite, grain boundary austenite, Widmanstatten austenite and secondary austenite.
2. By comparing the temperature fields of SDM and RM additive, with the increase of deposited layers, the heat accumulation of SDM additive is greater than that of RM additive, which can effectively verify the collapse phenomenon of SMD additive during the actual additive process, while the appearance of RM is well-formed.

3. The different paths of the additions also have different effects on the stress field. The simulation analysis shows that the stress at the bottom and top of the equivalent stress deposition layer are smaller and the middle stress is larger for different paths.

4. By comparing the longitudinal and transverse stresses along the incremental scanning direction and incremental height direction for different paths, the transverse stresses along the incremental scanning direction are both compressive stresses, and the SDM compressive stresses are greater than the RM compressive stresses. The longitudinal stresses along the increment height direction, near the top of the increment specimen, the tensile stresses of SDM are greater than the tensile stresses of RM.

**Author Contributions:** Methodology, Y.Y.; Formal analysis, J.G.; Data curation, X.B.; Writing—original draft, Y.Y. and C.J.; Writing—review and editing, R.L.; Funding acquisition, C.J. and R.L. All authors have read and agreed to the published version of the manuscript.

**Funding:** This work was supported by the National Natural Science Foundation of China (grant numbers 52075228).

**Institutional Review Board Statement:** Not applicable.

**Informed Consent Statement:** Not applicable.

**Data Availability Statement:** Not applicable.

**Conflicts of Interest:** The authors declare no conflict of interest.

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
