# Peer review of "Experimental and Numerical Investigation of CMT Wire and Arc Additive Manufacturing of 2205 Duplex Stainless Steel"

_coatings, doi:10.3390/coatings12121971_

Round 1

Reviewer 1 Report

1. The interlayer cooling time was 60s was maintained, but is the temperature is same or varied. Please report the interpass temperature of layers.

2. In Fig 2, the height of the deposition is gradually reduced towards the welding end point, what is the reason for variable height?

3. Is the model considered the cooling rate of the deposition? And how much temperature was used before starting the next layer? If the interlayer time is 60s, then the assumption of constant interpass temperature is not valid. Justify.

4. In fig 7, where is the location of the microstructures, is there any difference in microstructures at middle and top regions?

5. The variable temperature influences the grain growth and particular direction which influences the grain structure. Please refer the following articles and correlate the presented results accordingly.

1. Microstructure and Mechanical Properties of Hybrid-Manufactured Maraging Steel Component Using 4% Nitrogen Shielding Gas Fabricated by Wrought-Wire Arc Additive Manufacturing (Coatings).

2. Microstructural characteristics of wire arc additive manufacturing with Inconel 625 by super-TIG welding (TIIM)

Author Response

Comment 1The interlayer cooling time was 60s was maintained, but is the temperature is same or varied. Please report the interpass temperature of layers.

Response 1: Thanks for your suggestion. The interlayer cooling time of SDM and RM scanning paths is 60s seconds. By comparing the cooling temperatures of different paths, the highest temperature of the first layer cooled to 60s is 162℃. The highest temperature of the fifth layer cooling to 60s is 318℃ and 310℃, respectively. The highest temperature of the ninth layer cooling to 60s is 381℃and 367℃ respectively. RM is slightly faster than the SDM scanning path.

Comment 2In Fig 2, the height of the deposition is gradually reduced towards the welding end point, what is the reason for variable height?

Response 2: Thanks for your suggestion. The height of the deposition layer end of the SDM scanning path is reduced. It is mentioned in the manuscript.

When the heat sources move in a different way, the resulting temperature distribution inside the workpiece will also be different. When the heat source moves to the middle of the 10th layer, the temperature distribution of the left and right two-end points along the height direction is respectively taken,as shown in Fig. 11. The temperature of the arc leading end is lower than that of the arc extinguishing end. Comparing the temperature distribution between the arc-striking end and the arc-extinguishing end, it is found that the temperature of SDM is higher than that of RM at the arc-striking end, while the temperature distribution of SDM is consistent with that of RM at the arc-striking end. Therefore, in the process of SDM and RM additive manufacturing, SDM additive manufacturing will accumulate more heat, resulting in the process of one end collapse.

Comment 3Is the model considered the cooling rate of the deposition? And how much temperature was used before starting the next layer? If the interlayer time is 60s, then the assumption of constant interpass temperature is not valid. Justify.

Response 3: Thanks for your suggestion. The establishment of finite element model is based on the actual situation. In the process of modeling, factors such as scanning speed and time are considered. The interlayer temperature of each layer is changing. As the number of deposited layers increases, heat accumulation also increases.

Comment 4: In fig 7, where is the location of the microstructures, is there any difference in microstructures at middle and top regions?

Response 4: Thanks for your suggestion. Fig. 7 shows the top layer microstructure. I have added the microstructure at the bottom and middle part of the specimen.

By observing the multilayer single-pass samples along the BD, the microstructures at different positions can be observed as shown Fig.7. The bottom structure of the shape is mainly dendritic and feathery austenite. Widmanstatten austenite shows parallel arrangement, and amorphous grain boundary austenite content is more. The main microstructure in the middle is columnar dendrites and coarse widmanstatten austenite. The microstructure began to change from fine dendrite to columnar dendrite, and the grain size increased. At the top of the formed part, the microstructure is coarse widmanstatten austenite and columnar equiaxed grains, and the austenite size at the top position is larger than that at the middle position. This is because the heat dissipation condition at the top is the worst and the temperature gradient is the smallest, resulting in columnar dendrites.

Comment 5: The variable temperature influences the grain growth and particular direction which influences the grain structure. Please refer the following articles and correlate the presented results accordingly.

Response 5: Thanks for your suggestion. In the process of additive manufacturing, the grain growth size is affected by cooling rate and undercooling. At the junction of the substrate, the cooling rate is the fastest. In the process of additive manufacturing, the grain growth size is affected by cooling rate and undercooling. At the junction of the substrate, the cooling rate is the fastest. It can be seen that the bottom structure of the formed part is mainly dendritic and feathery austenite, Widmanstatten austenite is arranged in parallel, and the amorphous grain boundary austenite content is more. It can be seen that the main microstructure in the middle is columnar dendrites and coarse widmanstatten austenite. The microstructure began to change from fine dendrite to columnar dendrite, and the grain size increased. This is because with the repeated heating of the sample by the welding torch, the heat in the molten pool accumulates and is farther and farther away from the substrate. It can only rely on convection with air to dissipate heat, resulting in poor heat dissipation conditions in the molten pool. At the top of the formed part, the structure is coarse widmanstatten austenite and columnar equiaxed grains. The austenite size at the top is larger than that at the middle, because the heat dissipation condition at the top is the worst and the temperature gradient is the smallest.

Reviewer 2 Report

SUMMARY

This manuscript investigated the mechanical and thermal behaviour of 2205 duplex stainless steel produced by CMT wire and arc additive manufacturing. The publication contains an analysis of the influence of the direction of applying successive layers of material on the described material properties. The article's subject concerns the dynamically developing method of additive manufacturing of elements and may arouse interest, which justifies the article's publication. The paper is well structured and includes well-documented finite element analysis. However, some illustrations require corrections to improve readability. Therefore, it is recommended to make minor corrections before publication.

MAJOR COMMENTS

The abstract adequately introduces the topic of the article and contains critical information. The introduction is based on the literature from the last decade, which appropriately reflects the issues raised. The chapter on experimental methods contains clear descriptions and necessary illustrations. The quality of the descriptions of the pictures needs to be improved for readability. The strength of the article is the finite element analysis, expanding the area of knowledge concerning the distribution of temperatures and residual stresses during the implementation of the additive manufacturing process. The conclusions presented in the paper contain the most important observations from the conducted research.

 SUGGESTED IMPROVEMENTS

1.      Figure 1. Please introduce a more readable description.

2.      Figure 2. Please enter the designation of the two different additive scanning paths.

3.      Figure 6. Please introduce a more readable description.

4.      Line 238. Please explain how did you measure strains on such small samples.

5.      Figures 9 and 12. Please introduce a more readable description and enter the designation of the two different additive scanning paths.

Author Response

Comment 1: Figure 1. Please introduce a more readable description.

Response 1: Thanks for your suggestion. I described Figure 1 in detail.

Fig. 1 WAAM process. (a) Schematic diagram of CMT wire and arc additive manufacturing. (b) Extraction and size of tensile samples

Comment 2: Figure 2. Please enter the designation of the two different additive scanning paths.

Response 2: Thanks for your suggestion. I have modified Figure 2 as follows.

Fig. 2 Additive paths and macroscopic morphology. (a) SDM scanning path and macroscopic morphology. (b) RM scanning path and macroscopic morphology.

Comment 3: Figure 6. Please introduce a more readable description.

Response 3: Thanks for your suggestion. I have modified Figure 2 as follows.

Fig. 6 Mechanical properties. (a) Microhardness test. (b) Tensile test. (c-d) SEM fracture morphology diagram.

Comment 4: Line 238. Please explain how did you measure strains on such small samples.

Response 4: Thanks for your suggestion. This tensile test used CMT4303 microcomputer control electronic universal testing machine. Maximum tensile force is 30 KN. The stretched sample is small. The fixture during stretching was designed and the tensile test was carried out.

Comment 5: Figures 9 and 12. Please introduce a more readable description and enter the designation of the two different additive scanning paths.

Response 5: Thank you for your advice. I have modified Figures 9 and 12.

Fig. 9 Temperature field distribution at different midpoints for different paths. (a) SMD scan path. (b) RM scan path.

Fig. 12 Contour of the equivalent stress distribution for different paths. (a) SMD scan path. (b) RM scan path.

Round 2

Reviewer 1 Report

I appriciated the authors efforts on the revised manuscript to improve its quality.

Author Response

Comment 1:I appriciated the authors efforts on the revised manuscript to improve its quality.

Response 1: Thanks you for your appreciation. We will keep trying.